# Emerging Roles of Cullin-RING Ubiquitin Ligases in Cardiac Development

**DOI:** 10.3390/cells13030235

**Published:** 2024-01-26

**Authors:** Josue Zambrano-Carrasco, Jianqiu Zou, Wenjuan Wang, Xinghui Sun, Jie Li, Huabo Su

**Affiliations:** 1Vascular Biology Center, Medical College of Georgia, Augusta University, Augusta, GA 30912, USA; jzambranocarras@augusta.edu (J.Z.-C.); jzou@augusta.edu (J.Z.);; 2Department of Biochemistry, University of Nebraska-Lincoln, Lincoln, NE 68588, USA; xsun17@unl.edu; 3Department of Pharmacology and Toxicology, Medical College of Georgia, Augusta University, Augusta, GA 30912, USA

**Keywords:** E3 ubiquitin ligase, CRLs, heart development, cardiomyocyte maturation, ubiquitin, proteasomal degradation

## Abstract

Heart development is a spatiotemporally regulated process that extends from the embryonic phase to postnatal stages. Disruption of this highly orchestrated process can lead to congenital heart disease or predispose the heart to cardiomyopathy or heart failure. Consequently, gaining an in-depth understanding of the molecular mechanisms governing cardiac development holds considerable promise for the development of innovative therapies for various cardiac ailments. While significant progress in uncovering novel transcriptional and epigenetic regulators of heart development has been made, the exploration of post-translational mechanisms that influence this process has lagged. Culling-RING E3 ubiquitin ligases (CRLs), the largest family of ubiquitin ligases, control the ubiquitination and degradation of ~20% of intracellular proteins. Emerging evidence has uncovered the critical roles of CRLs in the regulation of a wide range of cellular, physiological, and pathological processes. In this review, we summarize current findings on the versatile regulation of cardiac morphogenesis and maturation by CRLs and present future perspectives to advance our comprehensive understanding of how CRLs govern cardiac developmental processes.

## 1. Introduction

The heart is among the first organs to form and function in mammals during development. The formation of a functionally competent heart capable of continuously supporting embryonic and postnatal life begins with a spatiotemporally regulated developmental process that involves the generation and subsequent contribution of different cardiac cell types [1,2]. Not surprisingly, errors during cardiogenesis can lead to congenital heart disease (CHD)—the most common birth defect, affecting approximately 1% of all liveborn infants [3]. Minor cardiac malformations, though often asymptomatic at birth, can be responsible for subsequent cardiac diseases in adulthood and among the aged population [4]. Therefore, understanding the mechanisms underlying cardiac development is fundamental to designing therapies that address CHD and other cardiac ailments. 

Cardiac development can be temporally divided into three stages: specification, morphogenesis, and maturation [5]. Specification refers to the cellular commitment of cardiac cell lineages from pluripotent progenitors derived from the mesoderm, a process that is primarily regulated by transduction of extracellular signals, including bone morphogenic proteins (BMPs), Wnt, fibroblast growth factor (FGF), and transforming growth factor-beta (TGFβ), which orchestrate the expression of transcription factors that establish specific cardiac genetic networks [6,7,8]. The specification of different cardiac lineages contributes to the formation of the first heart field (FHF) and second heart field (SHF), which give rise to the different heart chambers [6,9]. Cardiac morphogenesis comprises the steps leading to the appropriate spatial arrangement of cardiac cell lineages. At embryonic day 7.5 (E7.5) in mouse hearts, the FHF migrates to the heart-forming region and then extends to the midline, creating the cardiac crescent. Concurrent with this, the SHF localizes medial and anterior to the cardiac crescent. By E8.0, the heart tube forms in the ventral midline of the embryo. At this point, cardiac crescent cells migrate to the midline while SHF cells migrate posteriorly and anteriorly, forming a primitive beating heart tube. Structurally, the heart tube contains an outer layer composed of cardiomyocytes and an inner layer comprising endothelial cells; it also has an anteriorly positioned atrial pole (outflow) and a posterior venous pole (inflow). After attaining its shape at E8.5, the linear heart tube undergoes a rightward looping process driven by uneven growth and remodeling, resulting from the recruitment of cardiac cells differentiated from progenitor cells in the SHF [10]. This event leads to the formation of primitive atria and ventricles. The heart chambers are formed by E10.5, and during this phase of cardiac development, the venous pole moves anteriorly, setting the stage for further development of the cardiac chamber [2,6,8,9,11]. At approximately E9.5 and E10.5, cardiomyocytes undergo orientated cell division triggered by a series of signaling cascades from the endocardium and myocardium to form the trabecular myocardium that is outlined by a layer of endothelial cells, consisting of ridges within the ventricular wall [12,13,14,15,16]. The trabecular myocardium promotes nutrient exchange and motor force generation during the embryonic stage, when a coronary circulation is lacking [17]. Starting at E14.5, most of the trabeculae gradually regresses and fuses to the compact wall, a process termed compaction that results in the formation of a thick ventricular wall [12]. Cardiac maturation picks up at these late fetal stages and persists after birth [18]. During this transition stage, cardiomyocytes undergo a series of morphological, electrophysiological, and metabolic changes, a remodeling process that transitions cardiomyocytes from an immature to a mature state [5,17,19]. Major hallmarks of cardiomyocyte maturation include myofibril maturation, characterized by a more prominent sarcomere organization and switching of sarcomere proteins from fetal to adult isoforms. Electrophysiologic maturation encompasses an increased expression of ion channels (e.g., *Kir2.1*, *Kir2.2*, and *CACNA1C*), development of transversal-tubules (t-tubules), and redistribution of gap junctions from the periphery to the intercalated disc located at the ends of adult cardiomyocytes. These changes, among others, lead to more efficient electrical conduction throughout the myocardium. Central to metabolic maturation of cardiomyocytes are mitochondria. During maturation, mitochondria increase in size and number, undergo cristae maturation, and associate with sarcomeres [20]. This mitochondrial transformation is accompanied by isoform switching by some metabolic enzymes, including hexokinase (from HK1 to HK2), and the expression of enzymes responsible for fatty acid β-oxidation, changes that collectively result in a metabolic switch from the glycolytic preference of immature cardiomyocytes to the lipolytic metabolism favored by mature cardiomyocytes [5].

In recent decades, there has been intense interest in identifying transcriptional and epigenetic mechanisms that regulate cardiac development [2,5]. In contrast, relatively few studies have investigated post-translational mechanisms involved in regulating this process. A deep transcriptomic and proteome analysis revealed a significant mismatch between mRNA abundance and corresponding protein levels [21,22], underscoring the critical need to understand how alterations in the proteome affect cardiac development independent of transcriptomic changes. Furthermore, protein synthesis during cardiac development is a highly active process that has been shown to generate specific proteomes in developing cardiomyocytes compared with their adult counterparts, highlighting the essential role of protein regulation [5,23].

Among important protein-regulating processes is ubiquitin-mediated protein degradation, which is pivotal for the maintenance of protein homeostasis (proteostasis) in all cell types, including cardiomyocytes. Ubiquitin is a small protein modifier that can be covalently conjugated to protein substrates, a process termed ubiquitination. Ubiquitination of protein substrates is mediated by an E1-E2-E3 enzymatic cascade [24]. This is initiated by the activation of ubiquitin via a ubiquitin-activating enzyme (E1) in an ATP-dependent reaction. Activated ubiquitin is then transferred to a ubiquitin-conjugating enzyme (E2), followed by the covalent linkage of ubiquitin to lysine (Lys) residues in the target protein substrate mediated by a ubiquitin ligase (E3). Further ubiquitination cycles on ubiquitin itself (polyubiquitination) primarily targets substrates for degradation by the 26S proteasome [25].

The Cullin-RING E3 ligase (CRL) family is the largest family of E3 ubiquitin ligases, and its members are thought to degrade about 20% of the cellular proteome [26]. CRL complexes are formed from four components [27] (Figure 1): (1) cullin scaffold proteins, eight of which (CUL1, CUL2, CUL3, CUL4A, CUL4B, CUL5, CUL7, and CUL9) have been identified in the human genome; (2) a RING-box protein (RBX1 or RBX2); (3) adaptor proteins; and (4) a substrate-recognition receptor [27,28]. By associating with diverse substrate-recognition receptors, CRLs are capable of recognizing and degrading a myriad of substrates involved in pivotal cellular processes that regulate a wide spectrum of physiological and pathological events, including DNA replication and transcription, cell cycle progression, signal transduction, development, the circadian clock, protein quality control, stress responses, apoptosis, and viral modulation, among others [28,29] In this review, we summarize our current understanding of the roles of CRLs in cardiac development and discuss future perspectives highlighting research advances expected to provide deeper insights into CRL-mediated proteolysis in cardiomyocytes during cardiac development.

## 2. Neddylation of Cullins in Cardiac Development

The assembly and subsequent activity of CRLs depends upon the neddylation of the cullin protein [27,28], a process in which cullin proteins are covalently modified by the ubiquitin-like protein NEDD8 (neuronal precursor cell-expressed developmentally down-regulated protein 8) via a NEDD8-specific enzymatic cascade [30,31]. The neddylation of cullins occurs at conserved lysine residues in the C-terminal conserved region, usually adjacent to the cullin homology domain, and is aided by neddylation-specific E1 enzyme (NAE1-UBA3), E2 enzyme (UBE2M), and E3 ligases; neddylation, in turn, alters the conformation of cullins, stabilizing the structure of CRLs and facilitating the role of CRLs as E3 ubiquitin ligases [32].

NEDD8-activating enzyme 1 (NAE1) is the regulatory subunit of the only NEDD8 E1 enzyme [33]. The deletion of NAE1 in the heart via αMHC^Cre^ results in heart failure and, ultimately, neonatal lethality [34]. Mechanistically, it has been shown that the inactivation of CRLs by inhibiting neddylation leads to the accumulation of Hippo kinases, including MST1 (macrophage stimulating 1) and LATS1/2 (large tumor suppressor kinase 1 and 2), which phosphorylate YAP and inhibit its nuclear transport [35], preventing YAP-mediated cardiomyocyte proliferation. Consequently, the loss of NAE1 in the heart arrests cardiomyocyte proliferation and causes ventricular non-compaction, a phenotype characterized by excessive trabeculae and a thin myocardium compact layer. At molecular levels, it was shown that CUL7 mediates the ubiquitination and degradation of MST1 [34], whereas CUL4a controls the degradation of LATS1 [36]. Together, these findings suggest a pivotal role of CRLs in mid-to-late gestational ventricular chamber development and that the CRL-mediated proteolysis of Hippo kinases serves as a critical check point to activate YAP signaling and promote cardiomyocyte proliferation in the developing heart.

Consistent with these genetic studies, the acute pharmacological inhibition of neddylation in neonatal rat hearts with MLN4924, a potent neddylation inhibitor [26], leads to perinatal cardiomyopathies [37]. Despite the recovery of impaired cardiac function at 3 months of age, rats previously exposed to MLN4924 show exacerbated pathological cardiac remodeling and heart failure in response to the subsequent infusion of isoproterenol, indicating that the transient detrimental impact of MLN4924 on neonatal cardiomyocytes increases the susceptibility of the heart to subsequent cardiac stress [37]. This effect is due in part to the impaired proliferation of cardiomyocytes during cardiac development and maturation at the neonatal stage, which subsequently lead to a loss of cardiomyocyte numbers and compensatory cardiomyocyte hypertrophy. Whether this impairment is attributable to the prolonged disruption of CRL-dependent protein homeostasis remains to be explored. Moreover, because administration via intraperitoneal injection is an indirect and complicated systemic-delivery strategy, the impact of MLN4924 on non-cardiac cells and their contribution to cardiomyopathy under stress-overload conditions will require further evaluation. Despite these limitations, this study [37] highlights the importance of neddylation during cardiac development and suggests that its transient disruption during development can cause lifelong impairments in the heart.

Cardiac development continues after birth until cardiomyocyte maturation is complete. A recent report demonstrated the indispensable roles of neddylation in maintaining the integrity of the cardiomyocyte maturation process [38]. Specifically, this study showed that the mosaic deletion of NAE1 restricted to cardiomyocytes (induced by AAV9-cTnT-Cre injection into *Nae1*^f/f^ mice) disrupted multiple aspects of cardiomyocyte maturation, including t-tubule formation, cellular hypertrophy, fetal–adult isoform switching, and metabolic maturation (e.g., glycolysis to fatty acid oxidation transition), and subsequently led to cardiomyopathy [38]. Mechanistically, neddylation likely regulates cardiomyocyte metabolic maturation by controlling the levels of HIF1α (hypoxia-inducible factor 1-alpha) protein. During cardiomyocyte maturation, HIF1α plays a central role in mediating cardiomyocyte proliferation and metabolism and its expression is known to be primarily regulated by CUL2-VHL ubiquitin ligase [39,40]. Interestingly, this study reported that HIF1α is a putative NEDD8 target and further showed that the inhibition of neddylation and silencing of *Cul2* synergistically affected HIF1α protein levels [38], suggesting the intriguing possibility that neddylation regulates HIF1α through a CUL2-dependent and -independent mechanism [41,42,43].

Collectively, these observations suggest that neddylation fine tunes cardiac development throughout the entire process of embryonic and perinatal cardiac development. Given the indispensable role of neddylation in cardiac development and maturation, it would be valuable to further explore the specific mechanisms by which neddylation, whether CRL-dependent or -independent, regulates cardiomyocyte proliferation, metabolic maturation, and potentially other processes, such as the maturation of the cardiomyocyte cytoskeleton and Ca^2+^ handling.

## 3. Deneddylation of Cullins in Cardiac Development

Deneddylation, the process of removing NEDD8 from its target proteins, dynamically counterbalances neddylation. The COP9 signalosome (CSN), a protein complex containing eight subunits (CSN1–8), is primarily characterized as a deneddylation enzyme that removes NEDD8 from cullins [44,45]. All CSN subunits are needed to form the complex, with the deletion of any of the eight subunits impairing CRL activity [46]. The germline global deletion of *Csn2* [47], *Csn3* [48], *Csn5* [49], *Csn6* [50], or *Csn8* [51] was reported to cause embryonic lethality in mice.

Several studies have begun to uncover a pivotal role of CSNs in cardiac development and maturation. The depletion of CSN8 in mouse hearts via αMHC^Cre^ was shown to impair the activity of CRLs and lead to the accumulation of misfolded proteins in the heart. This, in turn, was shown to lead to increased cardiomyocyte necrosis and subsequent cardiac remodeling, heart failure, and premature lethality at approximately 4 weeks of age [44]. The pronounced cardiac remodeling observed in mutant hearts occurred between perinatal weeks 2 and 4, a time window during which cardiomyocytes undergo maturation involving a series of structural, metabolic, electrophysiological, and functional alterations, supporting an important role for CRLs in cardiac maturation.

Despite the demonstrated crucial role of CSN8 in post-mitotic cardiomyocytes [45], whether and how it might regulate cardiac development remains understudied. It was previously reported that the cardiac-restricted knockout of CSN8 results in elevated p62 and LC3-II protein levels and increased autolysosomes in cardiomyocytes, indicating the dual roles of CSN8 in regulating CRLs and autophagosome maturation [52]. Considering that cardiomyocyte development and maturation are tightly controlled by autophagy [53], determining whether and how CSN8 regulates cardiac development would be an exciting future research direction.

In addition to CSN8, another CSN subunit, CSN6, also has important roles in the heart and is potentially involved in cardiac development. A recent study reported that CSN6 is elevated in angiotensin II-induced hypertrophic mouse hearts and that the overexpression of CSN6 in neonatal rat cardiomyocytes induces hypertrophy, possibly by controlling the neddylation of CRLs and consequent regulation of SIRT2 protein levels [54]. It was also reported recently that either mutation of CSN6 in human patients or cardiac restricted knockout in transgenic mice causes disrupted neddylation-mediated protein degradation, accelerated desmosomal protein dissolution, and impaired desmosomal proteome degradation, which in turn lead to genetic-based desmosomal-targeted arrhythmogenic right ventricular (RV) dysplasia/cardiomyopathy (ARVD/C) [55]. These data together suggested an indispensable role of CSN6 in maintaining a healthy proteome in mature and developing hearts.

As one of the most important CRL regulatory mechanisms, neddylation is tightly controlled by the two-way neddylating–deneddylating process [56]. As discussed above, the absence of either NAE1 or CSN8 disrupts cardiac development and maturation, underscoring the importance of the precise regulation of neddylated CRLs in embryonic and perinatal hearts. The further exploration of the role of CSNs in cardiac development will strengthen the importance of CRLs in cardiac morphogenesis.

## 4. RBX1 and RBX2

The RING family proteins, RBX1 and RBX2, have dual functions in the regulation of CRL activity: mediating the transfer of activated ubiquitin from the E2 enzyme to the target substrate and promoting the neddylation of cullins. RBX1 associates with CUL1, CUL2, CUL3, CUL4A/B, and CUL7, whereas RBX2 is known to associate with CUL5. RBX1, a 14-kDa protein containing a RING finger domain (C3H2C3), is essential for the ubiquitination and degradation of a number of substrates involved in various biological processes [57,58]. A study employing a mouse gene-trap model demonstrated that the *Rbx1* gene is essential for early mouse embryogenesis and that its disruption causes early embryonic lethality at E7.5 [59]. The phenotype was linked to defective cell proliferation as a result of p27 accumulation. A recent study has uncovered the importance of RBX1 in cardiac wall morphogenesis in zebrafish [60]. It was shown that RBX1 deficiency caused hypertrabeculation and diastolic dysfunction in *Rbx1* mutant hearts. Interestingly, endocardial-specific, but not myocardial-specific, *Rbx1* overexpression rescued the cardiac wall phenotype [60]. To date, this function has only been demonstrated in the zebrafish model, and the role of RBX1 in the heart of any mammal remains to be determined.

RBX2, also known as SAG (sensitive to apoptosis gene), ROC2 (regulator of Cullin2), and RNF7 (RING finger protein 7), is a RING finger domain-containing protein that is ubiquitously expressed in many different organs and tissues but is prominently expressed in the heart, brain, and gonads [61]. Germline deletion of the *Rbx2* gene causes embryonic lethality at E11.5–12.5 in association with severe defects in vasculogenesis. The deletion of *Rbx2* also has a remarkable inhibitory effect on angiogenesis and the growth/proliferation of teratomas in vivo. Mechanistically, RBX2 is required for endothelial cell differentiation via a neurofibromin 1 (NF1)-dependent mechanism [62]. The endothelial deletion of *Rbx2* also causes embryonic lethality but at a later stage (E15.5), again with evidence of defective vasculogenesis and proliferation, indicating the crucial importance of RBX2 in vasculogenesis and, hence, embryonic viability [63]. Although RBX2 is highly expressed in the heart, its role in cardiac development has not been determined.

## 5. Cullin-RING Ubiquitin Ligases

### 5.1. CRL1

CRL1, also known as S-phase kinase-associated protein 1 (SKP1)-Cullin1-F-box (SCF) [29,64], consists of CUL1; RBX1; the adaptor protein, SKP1; and a variable F-box protein, which serves as a substrate-recognition receptor. CRL1 is the best-studied Cullin-RING E3 ligase and has most extensively been investigated as a regulator of the cell cycle; thus, research on CRL1 has focused on its role in cancer and therapeutic targets [65]. The importance of CRL1 in the cell cycle is showcased in a global *Cul1*-deficient model, which is characterized by early lethality during the gastrulation stage (E6.5), before the start of heart development, accompanied by accumulation of Cyclin E [66]. Some identified CRL1 targets involved in the cell cycle include p21, p27, Cyclin D, Cyclin E, and WEE1 [28]. CRL1 is expected to control cell cycle progression in the developing heart given the importance of proliferation and the dynamism of cell cycle proteins in this process [67].

F-box proteins, which confer substrate selectivity to CRL1, are known to be involved in diverse cellular events, including cell movement, metabolism, angiogenesis, cell death, DNA damage response, and—the most studied process—cell cycle progression [64]. A large number (>70) of F-box proteins have been identified. However, the physiologic roles of these F-box proteins in CRL1 remain to be discovered.

The F-box protein, FBX7, is known to be differentially expressed in the developing heart compared with the postnatal heart [68], but its functional importance in cardiac development is not known. FBXW7, another F-box protein, might be an important modulator of early cardiogenesis. Germline ablation of *Fbxw*7 in mice disrupted atrial ventricular chamber separation and maturation at E10.5, which was linked to an accumulation of Cyclin E and abnormal NOTCH signaling [69]. Subsequently, it was shown that FBXW7 targets NOTCH1 and NOTCH3 for proteasomal degradation [70].

FBXO32, a myocyte-specific F-box protein also known as Atrogin-1, has recently been studied in the heart in a zebrafish model. Knocking down *Fbxo32* in one-cell zebra fish embryos was shown to compromise heart function at 48 h post-fertilization in the absence of heart morphological abnormalities [71]. In a separate study, *Fbxo32* deletion in mice using a global knockout approach caused cardiac dysfunction only in aged hearts [72]. Both studies pointed to a role for FBXO32 in the regulation of autophagy. Despite these observations, the cardiac-specific functions of FBXO32 in mammalian hearts during development remain unknown. Additionally, FBXO25-associated CRL1 degrades the important cardiac-specific transcription factors, TBX5 and NKX2-5 [73]. Given the importance of these transcription factors in cardiomyocyte lineage commitment and the maintenance of cardiomyocyte identity, FBXO25 might be particularly important for cardiogenesis [5,74].

### 5.2. CRL2

CRL2 is composed of the scaffold protein CUL2, RBX1, two adaptor proteins (Elongin B and C), and a substrate-recognition receptor, with von Hippel–Lindau (VHL) being the most studied [28,68,75,76]. It is well known that CRL2 promotes the ubiquitination and subsequent proteasomal degradation of HIF1α [77,78]. Hypoxia-induced factors (HIFs) promote the expression of genes related to the cellular response to low oxygen levels. Specifically, HIFα (isoforms 1, 2, and 3) and HIFβ (also known as ARNT) form a heterodimer and translocate to the nucleus, where they act as transcription factors to control the expression of a wide spectrum of genes. HIF1β is constitutively expressed, whereas HIFα isoforms are oxygen-sensitive and, under normoxic conditions, are hydroxylated by prolyl hydroxylases (PHDs), which act as oxygen sensors. Hydroxylated HIFα subunits are then recognized by VHL, leading to their ubiquitination by CRL2 and subsequent proteasomal degradation. During hypoxia, the oxygen-dependent action of PHD is inhibited, allowing for HIFβ and HIFα subunits to associate and induce the transcription of HIF target genes [79].

A finely tuned balance of HIF1α is essential throughout cardiac development mainly due to its dual roles by controlling CM proliferation and energy metabolism. The cardiac development process occurs under hypoxic conditions [80]. On one hand, HIF1α plays an essential role in stimulating CM proliferation in the hypoxic embryonic hearts [39]. Meanwhile, this VHL loss-of-function approach (Nkx2.5^Cre^ Hif^f/f^) reported that HIF1α facilitates the modulation of ventricular chamber maturation by promoting metabolic transition [39]. This regulation of HIF1α is explained by the physiological events that take place during heart development. HIF1α exhibits sustained expression in the compact myocardium of the developing heart up to E14.5 [81]. As heart development proceeds and the coronary circulation is established in the fetal heart (~E14.5), the blood perfusion of the heart, and thus the oxygen level, increases, thereby repressing HIF1α signaling [82]. The observed metabolic transition, characterized by a shift towards a more glycolytic metabolism, is accompanied by the accumulation and hyperactivation of HIF1α and ventricular non-compaction in E14.5 hearts [81], highlighting the canonical role of HIF1α turnover in the developing heart. Furthermore, in support of a role for HIF1α in controlling metabolism, it has been reported that HIF1α promotes metabolic reprogramming during cardiac maturation by altering mitochondria size [83] and driving the transition from glycolysis to oxidative metabolism [38].

During this process, CRL2 is essential for the proper regulation of HIF1α and thus heart development and metabolic switching, but the specific mechanism by which this is accomplished is not completely understood. Another potential CRL2 target recognized by VHL is phospholamban (PLN), a protein the regulates intracellular levels of Ca^2+^ by modulating SERCA (sarcoplasmic/endoplasmic reticulum Ca^2+^-ATPase) activity. Despite the discovery that the degradation of phospholamban might be important during heart failure [84], the biological implication of this event for heart development has not yet been completely explored. However, it is possible to speculate on its importance given that SERCA expression is increased during the electrophysiological maturation that cardiomyocytes undergo [5]. In addition to VHL, CUL2 pairs with other CRL2 substrate receptors, such as FEM1 and KLHDC, but their targets in the heart have not been identified [68]. Therefore, the functional importance of these CRL2 substrate receptors in heart development remains to be established.

### 5.3. CRL3

CRL3 comprises CUL3, RBX1, and a group of substrate-receptor proteins. The best-characterized substrate-recognition adapters of CRL3, known as BBK (BTB-Back-Kelch) family proteins, contain an N-terminal BTB (bric a brac, tramtrack, and broad-complex BACK) domain and a C-terminal Kelch repeat domain [85]. The deletion or mutation of CUL3 or its adapter BTB proteins leads to a number of pathologies, including familial hyperkalemic hypertension [86], Gordon syndrome [87], diabetes mellitus [88], fibrotic kidney diseases [89], myoblast differentiation [90], and striated muscle diseases [91].

CLR3 is crucial for cardiac health. It has been reported that mice with the cardiac-restricted αMHC^Cre^-driven knockout of *Cul3* display a smaller body size, dilated right atrium and ventricle, severe cardiomyopathy, and atrial thrombi, followed by early neonatal lethality at P6 [91]. Immunofluorescence and proteomic analyses performed in this study further revealed disorganized and accumulated cTnT and an altered expression of proteins involved in oxidative phosphorylation and metabolic pathways, respectively [91]. The apparent myocyte impairment was not limited to the case of CUL3 depletion but was also observed with depletion of BTB domain-containing substrate-recognition adaptors. The mutation of the BTB domain in KLHL9 (Kelch-like family member 9), which disrupts KLHL9 binding to CLR3, was found to be associated with a unique form of early-onset autosomal-dominant distal myopathy in human patients [92]. The loss or mutation of the muscle-specific protein, KLHL40, in zebrafish, transgenic mice, or human patients leads to the degeneration of sarcomere thin filament proteins and severe nemaline myopathy [93,94]. In addition, it was reported that recessive small deletions and missense changes in *Klhl41* in zebrafish and *KLHL41* in humans are linked to diminished motor function, myofibrillar disorganization, and nemaline body formation, which are indicative of nemaline myopathy, whereas frame-shift mutations in the gene could lead to neonatal lethality [95]. Another mechanistic study in a transgenic mouse model revealed that KLHL41 is poly-ubiquitinated and potentially prevents against the aggregation and degradation of nebulin, an essential component of the sarcomere [96]. Furthermore, dominant mutant forms of KBTBD13 are associated with nemaline myopathy type 6 in human patients, characterized by the presence of nemaline rods and core lesions [97]. Proteomic analyses have suggested that KLHL13 recognizes the protein filamin C (FLNC), targeting it for CRL3 ubiquitin ligase-dependent degradation. KLHL13 deficiency in mice leads to congenital myopathies, including stunted postnatal skeletal muscle growth, centronuclear myopathy, Z-disc streaming, and sarcoplasmic reticulum dilation [98]. Taken together, these observations indicate that the CRL3 complex, including CUL3 and its BTB-domain–containing adapters, plays critical roles in fine-tuning myocyte development, likely by regulating cardiomyocyte metabolism and sarcomere and myofibril assembly, at least in part.

### 5.4. CRL4A/B

CRL4 can be formed from either of the two highly similar (83% sequence identity) scaffold proteins, CUL4A and CUL4B [99]. Additionally, CRL4 contains RBX1, the adaptor protein DDB1 (DNA damage binding protein 1), and a member of the DCAF (DDB1-CUL4 associated factor) substrate-receptor family [28,68]. Some identified CRL4 substrates include DNA damage-response proteins (e.g., CDT1 and XPC) as well as proteins associated with histone methylation (e.g., PR-Set7/Set8 and WDR5). The clinical relevance of CRL4 is exemplified by a reported mutation in *CUL4A* that leads to the development of cancer [100] and a mutation in *CUL4B* that results in X-linked retardation syndrome [101]. Despite the importance of CRL4 in human diseases and cellular events, its role in heart development has received little research attention.

There have been some studies on the role of CUL4A in cardiogenesis. In one such study evaluating the effects of *Cul4A* knockdown on zebrafish development [102], *Cul4A*-deficient zebrafish embryos exhibited pericardial edema and defective heart looping as well as alterations in their pectoral fin. The phenotypes were related to a reduction in proliferation and increase in apoptosis. It was proposed that CUL4A regulates the expression of *Tbx5* by binding directly to the *Tbx5* promoter. Nevertheless, whether this mechanism stands warrants further investigation. In contrast to results obtained in zebrafish, mice with a germline knockout of *Cul4A* develop normally, with males, but not females, showing mild cardiac dysfunction at 10 weeks of age [103].

Research has begun to uncover the significance of CUL4B in early embryogenesis, with different reports indicating that the global knockout of *Cul4B* leads to lethality at approximately E9.5 [104,105]. While the importance of CUL4B in proper blastocyst development is recognized, its role in cardiogenesis remains largely unexplored.

Interestingly, patients with limb-girdle muscular dystrophy (LGMD) present with severe cardiomyopathy and reduced muscle tissue levels of the CRL4 substrate-recognition receptor, DCAF6. The pathogenesis of LGMD remains elusive, but it is possible to speculate that the dysregulation of CRL4 in myocytes is involved [106]. Additionally, cardiac-specific *Dcaf6*-knockout (MCK^Cre^) mice show dilated hearts, cardiac dysfunction, and disrupted sarcomere structures at 12 weeks of age, a cardiac phenotype linked to disrupted F-actin binding and impaired mitochondrial function [107]. Moreover, the knockout of *Ddb1*, encoding an adapter that associates with CUL4 and recruits substrate-receptor proteins, results in embryonic lethality [27]. Overall, the roles of CUL4 in heart homeostasis and cardiac development remain poorly understood.

### 5.5. CRL5

CRL5 consists of CUL5, RBX2, the adaptor proteins Elongin B/C, and 1 of 37 substrate receptors [108,109]. CRL5 has been implicated in several biological processes and human cancers through its control over the turnover of a variety of substrates, but there are few reports on its role in cardiac development. One substrate of CRL5 is ASB2 (ankyrin repeat and SOCS box-containing 2), which is specifically expressed in the cardiac region as early as the appearance of the cardiac crescent (E7.5) in mice. The knockdown of *Asb2* disrupts normal cardiac development in zebrafish [110]. Interestingly, this cardiac phenotype is partially rescued by the simultaneous knockdown of the transcriptional regulator, *Smad9* [110]. Using *Asb2*-knockout embryos, it was further reported that ASB2 contributes to heart morphogenesis and function in mice by ubiquitinating filamin A [111]. To date, these are the only reported studies on the contribution of CRL5 to normal heart development. The specific involvement of CUL5 in cardiac development awaits the development of *Cul5*-knockout mouse models.

### 5.6. CRL7 and CRL9

CRL7 is composed of a non-canonical cullin, CUL7, which contains additional structural domains compared with CUL1–5, resulting in a higher molecular weight [27,68,112]. In addition to CUL7, the CLR7 complex also contains RBX1; the adaptor protein SKP1; and the substrate-recognition receptors, FBXW8 and FBXW11 [113].

A mutation in the *Cul7* gene is associated with 3M syndrome, an autosomal-recessive disease that causes growth and muscle abnormalities [114,115]. Interestingly, the germline deletion of *Cul7* has a global impact on growth, recapitulating the symptoms of 3M syndrome [115]. Mice lacking *Cul7* display reduced embryonic size relative to wild-type mice and show impaired placental vascular development, which leads to neonatal lethality [116]. The deletion of *Fbxw8* produces similar effects on size, but global *Fbxw8*-knockout embryos survive to adulthood despite their reduced size [117]. This milder phenotype compared with *Cul7*-deficient mice suggests that FBXW8 serves as one of the downstream effectors of CRL7 to regulate development. In the developing heart, CRL7 has been reported to mediate the degradation of MST1 [34], an important kinase in the Hippo–YAP signaling pathway, which is essential for heart development. This signaling cascade involves the sequential action of several adaptors and kinases that ultimately results in the phosphorylation of YAP (Yes-associated protein). In its unphosphorylated state, YAP translocates to the nucleus, where it serves as a transcriptional co-activator by associating with the transcription factors, TBX5, TEAD, and RUNX4, among others [8,118]. Thus, phosphorylation of YAP-mediated by the Hippo signaling pathway yields an inactive, cytoplasm-resident form of YAP. Extensive studies of the role of Hippo–YAP signaling during heart development using mice with the cardiac-specific knockout of different elements of the pathway have shown that Hippo–YAP signaling regulates organ size by controlling cell proliferation [119]. It is conceivable that the growth retardation characteristic of the 3M syndrome as well as global *Cul7*- and *Fbxw8*-knockout mice can be explained by the regulatory function of CRL7 in the heart and the Hippo–YAP signaling pathway; however, additional investigation will be required to confirm this.

Another well-documented CRL7 substrate is IRS1 (insulin receptor substrate 1), an adaptor protein that recruits kinases to insulin and IGF (insulin-like growth factor) signaling pathways [120]. Although IGF plays an important role in heart development by ensuring proper proliferation [121], it is not known whether IRS1 is involved during embryonic maturation because it has only been investigated in postnatal stages [122]. These uncertainties contribute to our poor understanding of the role of CRL7 in heart development.

CRL9 is the least-studied CRL. Its components and functions in cardiomyocytes and other resident cells are largely unknown. Current evidence indicates that CRL9 is important in maintaining nuclear and microtubule integrity by mediating the degradation of Survivin (BIRC5) [123]. The physiological roles of CRL9 in cardiac development await further investigation.

## 6. Concluding Remarks and Perspectives

Emerging evidence from studies targeting individual components of CRLs at different development stages have collectively demonstrated the significance of CRLs in cardiac development (Table 1). The power of CRLs to control heart development is attributable to the CRL-mediated proteolysis of a wide range of protein substrates involved in cardiomyocyte differentiation and proliferation, contractile apparatus assembly, metabolic transition, stress responses, and signal transduction (Figure 2). Despite these advances, our understanding of the specific involvement of CRLs in regulating cardiac development remains in its nascent stages. Additional research is needed to elucidate detailed mechanisms underlying the control of embryonic and postnatal cardiac development by CRLs and uncover the connections between CRL dysregulation and congenital and adult heart diseases. Insights into these fundamental questions hold the promise of preventing and treating cardiac diseases through targeting of CRLs. A number of specific research avenues warrant further investigation.

First, the clinical relevance of CRLs to cardiovascular diseases has not been well established. Genome-Wide Association Studies (GWAS) coupled with powerful bioinformatic analyses can be performed to identify associations of single-nucleotide polymorphisms (SNPs) of any CRL component, especially diverse substrate receptors, with CHDs and other cardiomyopathies. In silico analyses of available bulk and single-cell RNA sequencing datasets from hearts with CHD may reveal the potential involvement of CRL dysregulation in the pathogenesis of these diseases. Importantly, changes in transcript levels may not always translate directly into changes in protein levels. Thus, it is critical to assess the expression of CRL subunits in diseased hearts at the protein level.

Second, the physiological importance of CRLs in governing different aspects of cardiac development remains to be convincingly demonstrated. Specifically, little is known about whether and how individual CRLs regulate cardiac looping, outflow tract formation, cardiac chamber specification and separation, trabeculation and compaction, and cardiac maturation. Genetically modified mouse models generated by targeting individual components of CRLs will be instrumental in addressing these questions. Such models represent powerful tools for modulating the activities of CRLs at different developmental stages in specific types of cardiac-resident cells, including cardiomyocytes, endothelial cells, fibroblasts, smooth muscle cells, epicardial cells, macrophages, lymphatic endothelial cells, and even neural cells, all of which are crucial for the formation of a functional adult heart. Addressing the specific roles of CRLs in these cell types in cardiac development represents a particularly intriguing challenge. With technological advancements in the field, it should prove possible to use human-induced pluripotent stem cell-derived cardiomyocytes to study how CRLs regulate cardiomyocyte differentiation and maturation. Findings obtained using this superior platform may provide new insights not only into cardiac development but also regarding myocardial regeneration.

Third, research efforts should be directed to understanding the molecular mechanisms by which CRLs regulate cardiomyocyte function. Since cardiomyocytes may have a unique proteome compared with other cell types, it is important to define the proteome regulated by CRLs in cardiomyocytes and hearts using a proteomics approach. In this context, ubiquitin remnant enrichment coupled with quantitative multiplex proteomics [124,125] can be used to identify specific substrates of CRLs in cardiomyocytes. Proteins that interact with individual CRLs could include substrates, substrate receptors, and ancillary proteins that regulate CRL activity. Identification of the interaction partners of CRLs, for example, by taking advantage of the continuously evolving proximity biotinylation approach [126], may define cardiomyocyte-enriched or -specific substrate receptors, substrates, and regulatory proteins of CRLs. Notably, it has been estimated that the human genome encodes more than 400 CRL substrate receptors [127,128]. To date, the functional importance and the substrates of most of these CRLs in cardiomyocytes are largely unknown. While the majority of past and ongoing studies of CRLs have concentrated on their proteolytic function, it is possible that CRLs regulate types of ubiquitination that do not lead to protein degradation, such as mono- and lysine 63-linked ubiquitination. Determining whether CRLs mediate the non-proteolytic ubiquitination of protein substrates in cardiomyocytes and establishing the significance of such non-proteolytic ubiquitination in the regulation of cardiomyocyte function and cardiac development represent interesting avenues for future research. Finally, despite a growing appreciation of CRLs in health and disease, the mechanisms that regulate CRL activity are not fully understood, especially in cardiomyocytes. Evidence, mostly obtained from cultured cells, has consistently pointed to the necessity of cullin neddylation in the regulation of CRL assembly and activity [127]. However, the pathophysiological significance of CRLs in different tissues has mostly been inferred from findings in animals harboring deletions of individual cullins, and definitive evidence of the importance of cullin neddylation in vivo is largely lacking. Not-yet-generated neddylation-deficient cullin-knockin mouse models, in which the lysine-accepting NEDD8 is conditionally mutated to arginine in a Cre-dependent manner, will shed light on this unanswered question. On the basis of findings from cultured cells, it has been proposed that the neddylation of cullins is mediated by DCN-type NEDD8 E3 ligases (DCUN1D1-DCUN1D5) [129]. Whether any of these proteins are necessary and/or sufficient to mediate cullin neddylation in cardiomyocytes in vivo is not known. Additionally, whether cellular and pathological stresses and/or environmental cues, such as hormones and neuronal signals, impact CRL activities remains an open question. If so, it will be interesting to understand how these factors affect CRL activities, for instance, by controlling the neddylation of cullin or post-translational modifications of CRL components. Findings from these lines of investigation hold the promise of preventing and treating cardiac diseases by targeting CRLs.

## Figures and Tables

**Figure 1 cells-13-00235-f001:**
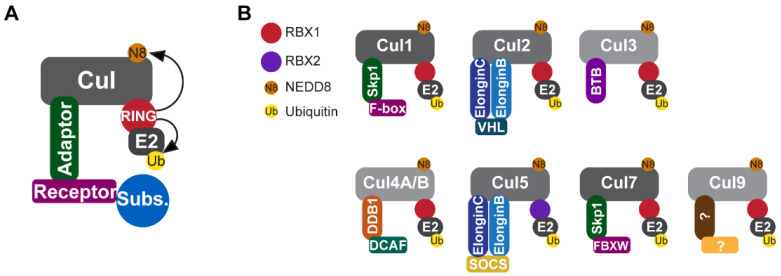
Components of CRLE3 ligases. (**A**) CRLs are composed of a cullin scaffold protein, a RING-box protein (RBX1 or RBX2), an adaptor protein, and a substrate-recognition receptor. RBX1 or RBX2 activates the CRL complex by promoting the neddylation of cullins and recruiting the ubiquitin-loaded E2 to the CRL complex. (**B**) All CRLs (CRL1–9) possess the same basic structure, but differ in the components that recognize diverse substrates. Note that CLR5 is the only known CRL that associates with RBX2. CRL, Culling-RING E3 ligase; RBX1, RING-box protein 1; RBX2, RING-box protein 2.

**Figure 2 cells-13-00235-f002:**
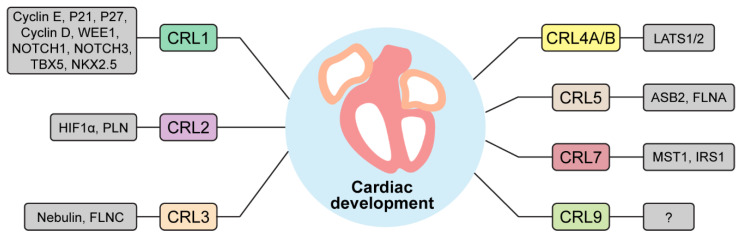
A list of CRLs and their targets important for heart development. CRL1-9 are depicted with their respective substrates. WEE1, WEE1 G2 checkpoint kinase; NOTCH1-3, neurogenic locus notch homolog protein 1-3; TBX5, T-box protein 5; NKX2.5, homeobox protein NK-2 homolog E; HIF1α, hypoxia inducible factor 1α; PLN, phospholamban; FLNC, filamin C; LATS1/2, large tumor suppressor kinase 1/2; ASB2, ankyrin repeat and SOCS box containing 2; FLNA, filamin A; MST1, mammalian Ste20-like kinase 1; IRS1, insulin receptor substrate 1.

**Table 1 cells-13-00235-t001:** Cardiac phenotypes of individual knockout mice.

Gene	Strategy	Cardiac Phenotype	Mechanism	Ref.
*Nae1*	αMHC^Cre^, mice	Ventricular non-compaction (E16.5), cardiac dysfunction (P1), and neonatal lethality (P3-7)	Accumulation of MST1 and LATS1/2; Hippo-YAP pathway inactivation	[34]
*Csn8*	αMHC^Cre^, mice	Necrosis and cardiac remodeling, heart failure, and premature lethality	Elevated p62 and LC3-II; impaired autophagosomes through regulation of Rab7 expression	[44]
*Csn8*	αMHC-Mer^Cre^-Mer, mice	Necrosis and cardiac dysfunction	Impaired autophagosome removal and elevated oxidized protein levels	[45]
*Rbx1*	Mouse gene trap model	Embryonic lethality (E7.5)	Proliferation defects due to p27 accumulation	[59]
*Rbx1*	Tie2^Cre^, zebrafish	Dysregulated cardiac wall morphogenesis	Modulation of GLI1 levels in the endocardium	[60]
*Rbx2*	Mouse gene trap model	Embryonic lethality (E11.5-12.5)	Defective vasculogenesis through accumulation of NF1 and Ras inhibition	[62]
*Rbx2*	Tie2^Cre^, mouse	Embryonic lethality (E15.5)	Defective vasculogenesis and proliferation	[63]
*Cul1*	Mouse gene trap model	Embryonic lethality (E6.5) during gastrulation	Accumulation of cyclin E	[66]
*Cul3*	αMHC^Cre^, mouse	Smaller body size, dilated right atrium and ventricle, severe cardiomyopathy, and atrial thrombi, followed by early neonatal lethality at P6	Dysregulation of cardiac anti-oxidative and metabolic processes	[91]
*Klhl40*	Mouse gene trap model	Severe nemaline myopathy	Loss of sarcomere thin filament proteins	[94]
*Klhl41*	Antisense morpholinos, zebrafish	Neonatal lethality	Defective motor function and myofibrillar disorganization with nemaline body formation	[95]
*Klhl41*	Mouse gene trap model	Nemaline myopathy	Disruption of sarcomeres and aberrant expression of muscle structural and contractile proteins resulting from prevention of nebulin aggregation and degradation	[96]
*Klhl13*	CRISPR-Cas9 gene editing, mouse	Congenital myopathies	FLNC upregulation	[98]
*Asb2*	Antisense morpholinos, zebrafish	Abnormal cardiac development	Regulation of *Tbx2* expression through SMAD9 degradation	[110]
*Asb2*	E2a^Cre^VEC-Cre, mouse	Early death due to heartbeat defects	Regulation of FLNA degradation in immature cardiomyocytes at the onset of myofibrillogenesis	[111]

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
