# Peer review of "Emerging Roles of Cullin-RING Ubiquitin Ligases in Cardiac Development"

_cells, 2024, doi:10.3390/cells13030235_

Round 1

Reviewer 1 Report

Comments and Suggestions for Authors

The review article by Zambrano-Carrasco et al. provides a comprehensive summary of current findings regarding the versatile regulation of cardiac morphogenesis and maturation by CRLs. The authors also discuss future perspectives that could advance our understanding of how CRLs govern cardiac developmental processes. Overall, it is a well-written and well-structured review that provides the latest advances in the study of CRL regulation of cardiac morphogenesis and maturation. Nonetheless, I have two minor suggestions. For the statement, "Once the linear heart tube has attained its shape (E8.5), it undergoes a rightward looping process driven by the uneven growth and remodeling caused by the recruitment of cardiac progenitor cells from external regions of the developing heart to the arterial and venous poles”, it should note that the rightward looping process is driven by the uneven growth and remodeling caused by the recruitment of cardiac cells differentiated from SHF progenitor cells. The authors also stated that “At approximately E9.5 and E10.5, cardiomyocytes invaginate to form the trabecular myocardium”, however, latest studies demonstrate that cardiac trabeculae form by directional cardiomyocyte migration and oriented cell division rather than invagination (1: Liu J et al., A dual role for ErbB2 signaling in cardiac trabeculation. Development. 2010 Nov;137(22):3867-75. doi: 10.1242/dev.053736, and 2: ). Please revise both statements accordingly.  

Author Response

Please see the attachment, thanks.

Reviewer 2 Report

Comments and Suggestions for Authors

In their review, the authors describe the importance of Cullin-RING ubiquitin ligases in cardiac development. The manuscript provides a very in-dept look on this important issue. The review is well written and very comprehensive. Maybe, providing another illustration of effects regulated by Cullin-RING ubiquitin ligases would be helpfull, or e.g. a graphical abstract.

Author Response

Please see the attachment, thanks.

Reviewer 3 Report

Comments and Suggestions for Authors

The authors summarize current findings on the versatile regulation of cardiac morphogenesis and maturation by CRLs and present future perspectives to advance our comprehensive understanding of how CRLs govern cardiac developmental processes. Overall, the review is well organized with a certain level of innovation and foresight. However, there are still some questions need to be addressed:

1.       It is better to have a figure showing the roles of CRLs with their substrates in the regulation of cardiac development.

2.      The references need to be adjusted with more recently published papers, only one reference published last year was cited in the list.

Author Response

Please see the attachment, thanks.
